# Research on high performance combustible gas concentration sensor based on pyramid beam splitter matrix

**Boqiang Wang** [1,2] *, **Xuezeng Zhao**[1], **Yiyong Zhang**[1,2], **Zhuogang Wang**[1]

**1** Harbin Institute of Technology, Harbin, Heilongjiang, China, **2** China State Shipbuilding Corporation Limited 703 Research Institute, Harbin, Heilongjiang, China

* 2534786721@qq.com

**Data Availability Statement:** All relevant data are within the manuscript and its Supporting Information files.

**Funding:** The author(s) received no specific funding for this work.

## Abstract

Combustible gas concentration detection faces challenges of increasing accuracy, and sensitivity, as well as high reliability in harsh using environments. The special design of the optical path structure of the sensitive element provides an opportunity to improve combustible gas concentration detection. In this study, the optical path structure of the sensitive element was newly designed based on the Pyramidal beam splitter matrix. The infrared light source was modulated by multi-frequency point signal superimposed modulation technology. At the same time, concentration detection results and confidence levels were calculated using the 4-channel combustible gas concentration detection algorithm based on spectral refinement. Through experiment, it is found that the sensor enables full-range measurement of CH4, at the lower explosive limit (LEL, CH4 LEL of 5%), the reliability level is 0.01 parts-per-million (PPM), and the sensor sensitivity is up to 0.5PPM. The sensor is still capable of achieving PPM-level detections, under extreme conditions in which the sensor's optical window is covered by 2/3, and humidity is 85% or dust concentration is 100mg/m$^3$. Those improve the sensitivity, robustness, reliability, and accuracy of the sensor.

## Introduction

It is a threat that fires and even explosions caused by combustible gas leaks to safety production in the petrochemical field, which accounts for about 50% of all major petrochemical accidents in China [1–3]. Although combustible gas sensors are prevalent in the workplace and have alarmed considerable staff for emergency response, about 40% of flammable and explosive accidents in petrochemical production sites were caused by no or failed alarms [4,5]. On the other hand, there were over 300,00 alarms in China responded to by fire departments in 2022 due to sensor malfunctions, leading to disruption to normal production, spending of social resources, and decreasing public confidence [6,7]. Prompt and reliable combustible gas leaks detection is critical for rescuing life and avoiding flammable and explosive damage.

Non-dispersive infrared (NDIR) is the most widely used sensor for combustible gas concentration. Fig 1 shows a schematic diagram of a simple NDIR gas sensor. Typically, emission

**Competing interests:** NO authors have competing interests

from a broadband source is passed through two filters, one covering the whole absorption band of the target gas (in the active channel), and the other covering a neighboring non-absorbed region (the reference channel). Provided that the chosen active and reference channel filters do not overlap significantly with the absorption bands of other gas species present in the application, cross-sensitivity to other gases lies below the limit of detection. Detects the concentration of combustible gas by the degree of absorption of a light source.

Combustible gas concentration detection technology faces challenges of improving detection limits, false alarm resistance, and adaptation under extreme using environments. In the very early stages of a fire, a sensor with at least 3 PPM sensitivity and 1 PPM accuracy is required [8]. It is a key instrument that increases the optical path length, such as reflective optics [9] and compact pentahedron reflector structures [10] that increase the light path length through the reflection process. In addition to that, detection limits can be improved by increasing the efficiency of the sensitive element, for instance, plating zirconate titanate (PZT) film on the surface of the sensitive element [11], and etching 3D patterns on the Monocrystalline lithium tantalate film [12]. Furthermore, improving the infrared light source efficiency is another way to improve detection limits, for example, installing high refractive index long period gratings (LPFG) in front of the infrared light source [13], and improving the infrared light source structure based on Fabry-Borot (FPG) structure [14]. All the methods above can achieve sub-ppm level detection. However, at the very beginning of the leak, the concentration of combustible gases is below the PPM level, at the same time, the calculation results need to be evaluated at extremely lower concentrations. In more serious cases, the reliability of the sensor will be affected when the window of the sensor is attached by contaminants, or the infrared detection channel is aging or malfunctioning.

The previous studies did not have a precise design of the optical path structure, which resulted in the infrared light not being sufficiently absorbed by the combustible gas in the tiny sensor optical path structure. Thus, the sensitivity and accuracy of the sensor could not meet the requirements. The pyramid beam splitter creates multiple reflections in the tiny sensor optical path structure, greatly increasing the effective optical range and allowing infrared light to be fully absorbed by combustible gases. In this study, a 4 channels infra-red-sensitive element with a new optical structure is designed to improve sensor accuracy, sensitivity, and reliability. Pyramidal beam splitter matrix is used as an optical re-flection structure inside the sensitive element. On the one hand, the sensitivity of the sensor is improved by increasing the optical path length. On the other hand, the complex optical re-flection path enables the 4 signal detection channel to receive a uniform infrared light signal, and effectively reduces the impact of contaminants (such as condensation fog, dust, etc.) attached to the optical window on the performance of the sensor, thus raising the reliability of the sensor. The concentration calculation results and the confidence level are calculated by the results of the 4 infrared signal detections that are calculated by 4 channel combustible gas concentration redundancy algorithm based on the linear frequency modulation chirp z-transform (CZT). The designed combustible gas concentration sensors are tested for detection performance, accuracy and sensitivity, and reliability, by the combustible gas concentration calibration experiment, the combustible gas limit detection experiment, and the anti-interference capability simulation experiment, respectively.

## 2. Design of sensitive element and measurement circuit

### 2.1 Sensitive element design

In order to increase the optical length and mix infrared light well, as shown in Fig 2, a pyramid beam splitter matrix is added in the center of the sensitive element. At the same time, the

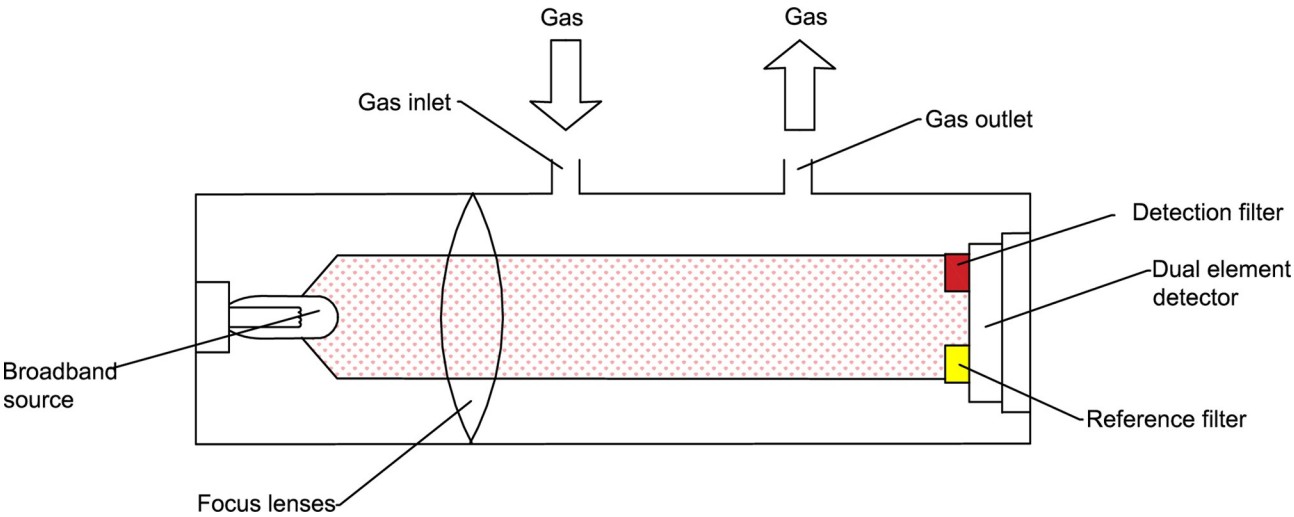

**Fig 1. Schematic structure of NDIR combustible gas concentration sensor.**

combination of 4 infrared bandpass filters and infrared pyroelectric elements is placed around the pyramidal beam splitter matrix. An optical incidence window is arranged at the top of the sensitive element, with an infrared reflective lining on the inside. The infrared reflective lining has high pass-through to infrared light signals from the entrance to the inside of the sensitive element, and it has full-spectrum high reflectivity for infrared light signals reflected by the pyramidal spectral matrix and infrared bandpass filter.

The top view of the inside of the sensitive element is shown in Fig 3. An infrared pyroelectric sensitive element is arranged behind each group of infrared bandpass filters (A1, A2, B1, B2). A1 and A2 only pass through the infrared light signal with a center wavelength of 3.4$\mu$m and a bandwidth of 0.2$\mu$m, for the detection of combustible gas. B1 and B2 only pass through the infrared light signal with a center wavelength of 3.91$\mu$m and a bandwidth of 0.2$\mu$m, to provide a reference for the detection waveform. A1, A2, B1, and B2 have full-spectrum high reflectivity for infrared light signals outside the allowed passage wavelength.

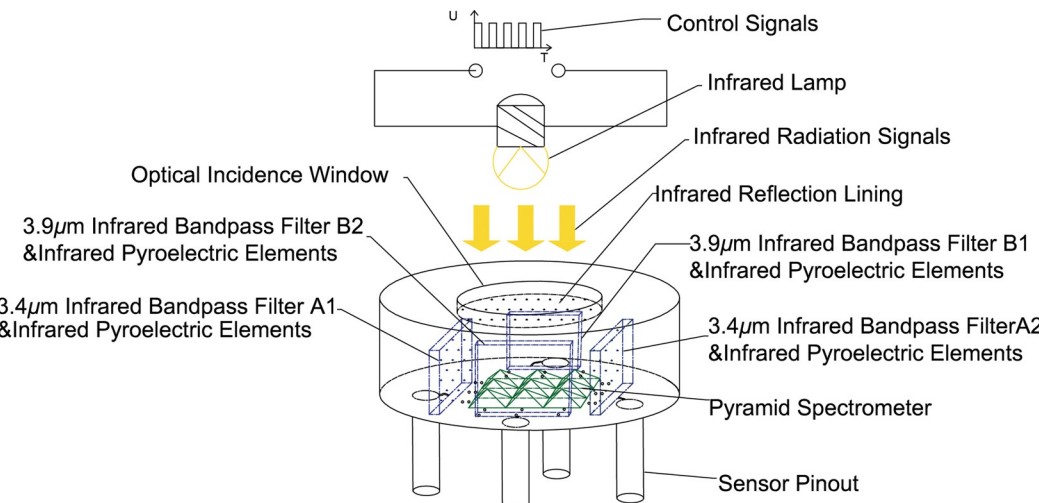

**Fig 2. Schematic diagram of the structure of the sensitive element.**

As shown in Fig 4, Infrared light A and B, to be emitted by the infrared light source, are shot inside the sensitive element through the signal inlet on it. The infrared light is reflected toward the infrared bandpass filter when it hits the pyramidal beam splitter. Only the infrared light, within the bandpass wavelength range, can pass through it and shot to the infrared pyroelectric element behind it. The rest of the infrared light, out of the bandpass wavelength range, will all be reflected into the infrared reflection lining, and a secondary reflection - all the infrared light will be reflected, due to its highly reflective full spectrum - occurs here. The secondary reflected infrared light will be reflected to the bandpass filter in the opposite direction, and repeat the above optical reflection process.

In Fig 4, only the reflection process of two beams of light (A and B) is shown between the two infrared bandpass filters and infrared reflection lining. Actually, it is a complex reflection process that a beam of light is reflected between the pyramid beam splitter matrix, 4 infrared bandpass filters (A1, A2, B1, and B2), and the infrared reflection lining. This process can increase the optical path length and provides all infrared light signals in the respective wavelength range to 4 infrared bandpass filters (A1, A2, B1, and B2).

As mentioned above, this design for the sensitive element can improve the reliability and accuracy of the combustible gas concentration sensor.

On the one hand, discuss the impact of improving the accuracy of the sensor for the detection of combustible gas concentration. Absorption properties follow the Beer-Lambert law [14,15]:

$$I = I_0 e^{-\alpha C L} \tag{1}$$

Where $I_0$ is the intensity of the infrared light incident, $I$ is the intensity of the infrared light transmitted through the gas, $\alpha$ is the absorption coefficient of the gas, $L$ is the optical path length, and $C$ is the concentration of the gas. Apparently, increasing the optical path length can improve the absorption of infrared light by the combustible gas, when the concentration of the combustible gas is certain, and improve the accuracy of the sensitive element to detect combustible gas concentration. The complex optical reflection process in the internal optical path structure of the sensitive element can increase the optical path length so that the infrared light signal can be fully absorbed by the combustible gas. The electrical signal difference can be

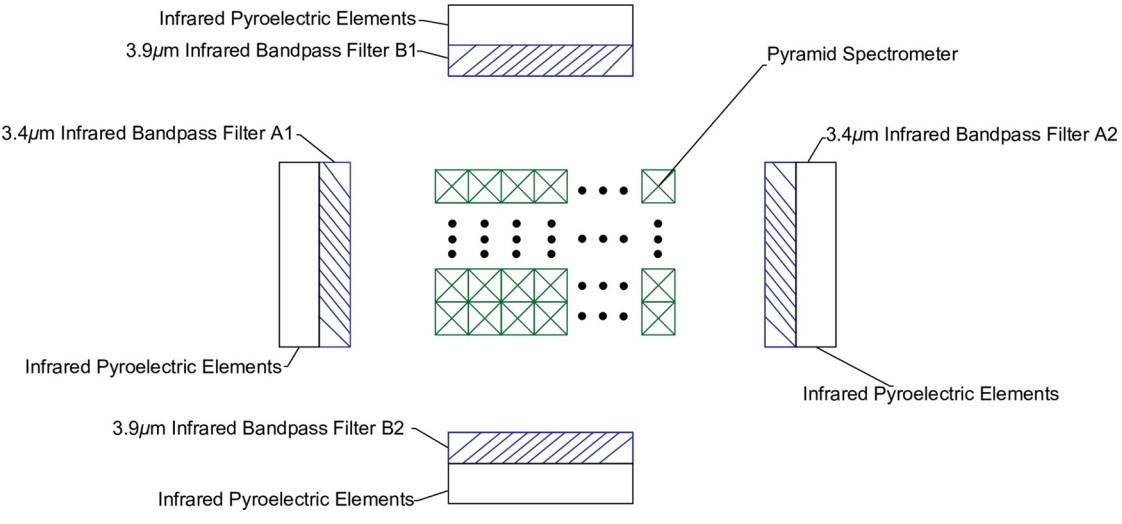

**Fig 3. Top view of the inside of a sensitive element.**

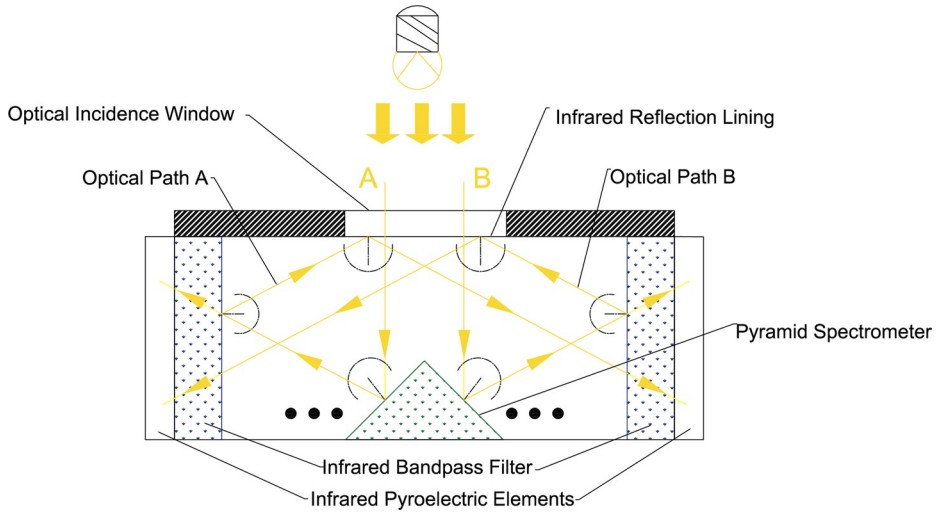

**Fig 4. Schematic diagram of the optical path reflected by the pyramid beam splitter matrix.**

increased between the detection channel and the reference channel, thus increasing the accuracy of the sensor for the detection of combustible gas concentration. The electrical signal difference can be adjusted by adjusting the position of the digital potentiometer Wiper U25 in Fig 5.

On the other hand, discuss the impact of improving the reliability of the sensor. The condensation fog can be formed on the optical incidence window of the sensitive element, due to the fact that the sensor is affected by environmental temperature factors (such as the temperature difference between day and night, or the system between start and stop, etc.). And dust particles are inevitably attached to the optical incidence window of the sensitive element, on account of that the sensor inlet and outlet are open paths. All of these conditions can lead to the varying reduction of infrared signals that are received by each of the infrared signal detection channels, so that the detection malfunction occurs. The complex optical reflection process inside the sensitive element enables each of the infrared signal detection channels to receive a

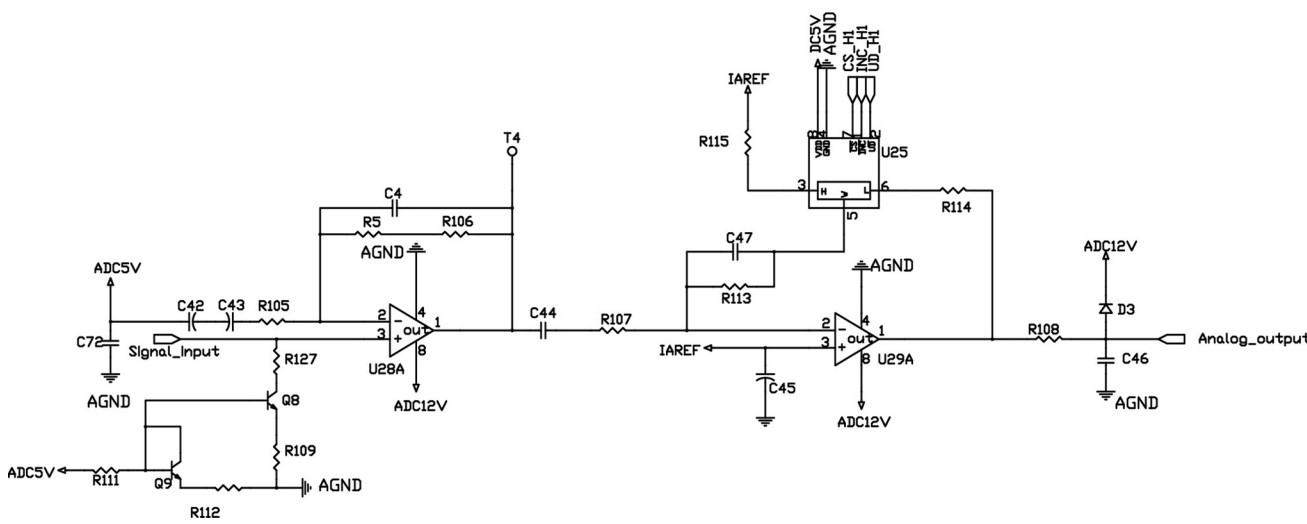

**Fig 5. Schematic diagram of the measurement circuit.**

uniform signal. Even if the incidence window is attached to contaminants–it causes non-uniform incidence of infrared light signal into the sensitive element - each of the infrared signal detection channels can receive the uniform signal, too. Through the reflection of the pyramid reflector matrix in the internal optical structure of the sensor, the infrared light can be optically mixed, so that each channel receives the same infrared light composition. This process also increases the effective optical range, so that the sensor's sensitivity and detection accuracy are not affected by the adherence of contaminants.

It thereby ensures that the sensor can effectively detect the concentration of combustible gases even under unfavorable conditions, thus increasing the reliability of the sensor.

## 2.2 Measurement circuit design

The circuits of measuring signal from four detection channels of the sensitive elements are completely identical. This circuit diagram is shown in Fig 5. Therefore, only one of the detection circuits is used as an example. The circuit consists of 2 operational amplifiers (op. amps.) U28A and U29A, 2 transistors Q8 and Q9, and a digital potentiometer U25. The very-low amplitude raw output at the sensitive element is amplified through the two-stage amplifier circuit that is constituted by U28A and U29A. And these two transistors, Q8 and Q9 behave as a temperature compensation circuit that can suppress the temperature drift of the output signal from the sensitive component. The analog signal is precisely regulated by the digital potentiometer U25. Finally, the processed analog signal is passed to the analog-to-digital converters (ADC). The calculation of combustible gas concentration is discussed in section 3.

# 3. Combustible gas concentration detection algorithm based on CZT principle

## 3.1 CZT algorithm principle

CZT algorithm is often used for spectrum refinement in the characteristic bands of the signal, and it has the advantages of flexible refinement scale and high accuracy [16].

The specific calculation process can be expressed as follows: Suppose signal sequences of limited length $x(n)$ are spectrally refined in the frequency band with the origin frequency $f_0$, end frequency $f_L$ and bandwidth length $M$, $M = f_0 + f_L$. It is done by CZT transformation [17–22].

$$X(Z_r) = CZT[x(n)] = \sum_{n=0}^{M-1} x(n)A_0^{-n}e^{-j\theta_0 n}w_0^{-nr}e^{-j\varphi_0 nr} \tag{2}$$

where $\theta_0$ is the initial amplitude angle, $\varphi_0$ is equally spaced increments on a unit circle angle, $A_0$ is the length of vector radius at the starting sampling point, $w_0$ is the elongation of the Z-plane helix, $j$ is the imaginary number, and the superscript $r$ denotes the serial number. Sample on the unit circle, when $A_0$ and $w_0$ are equal to 1 at the same time. We can derive Eq (3) from Eq (2).

$$X(r) = \sum_{n=0}^{M-1} x(n)\exp[-j(\theta_0 + \varphi_0 r)n] \tag{3}$$

where $\theta_0$ is equal to $2\pi f_0/f_s$, $\varphi_0$ is equal to $2\pi f_L/(Mf_s)$ and $f_s$ is the sampling frequency of the signal. Therefore, the frequency resolution of this band $\Delta f$ after refinement of the analysis is equal to $f_L/M$.

## 3.2 Four-channel combustible gas concentration redundancy calculation model

As shown in Fig 6, The drive signal of the infrared light is driven by the modulated waveform which is superimposed by the sinusoidal signal with a frequency of 4.0–5.0Hz and an interval of 0.1Hz.

In Fig 5, the characteristic frequency of the signal output from the secondary amplifier circuit is between 4 and 5 Hz, with an interval of 0.1 Hz. Consequently, this band (between 4 and 5 Hz) is the characteristic frequency band of the signal, and with the frequency refinement scale of 0.1 Hz, refine the spectrum for the characteristic band using the CZT algorithm.

Suppose that the modulus sums of the 4 secondary amplified signals calculated by the CZT algorithm are $M_1$, $M_2$, $R_1$ and $R_2$ on a minimum resolution of 0.1 Hz in the signal characteristic frequency band between 4 and 5 Hz. Where $M_1$ and $M_2$ are respectively the mode sums of the infrared pyroelectric element 3.4μm band channel 1 and channel 2, $R_1$ and $R_2$ are respectively the modulus sums of the infrared pyroelectric element 3.91μm band channel 1 and channel 2. Then, the expression can be written as follows:

$$
\begin{cases}
M_1 = \sum_{Z_r=4.0}^{5.0} X(Z_r) = \sum_{Z_r=4.0}^{5.0} \sum_{n=0}^{M-1} x(ADC_1)\exp[-j(\frac{2\pi f_0}{f_s} + \frac{2\pi f_L}{Mf_s}r)n] \\
M_2 = \sum_{Z_r=4.0}^{5.0} X(Z_r) = \sum_{Z_r=4.0}^{5.0} \sum_{n=0}^{M-1} x(ADC_2)\exp[-j(\frac{2\pi f_0}{f_s} + \frac{2\pi f_L}{Mf_s}r)n] \\
R_1 = \sum_{Z_r=4.0}^{5.0} X(Z_r) = \sum_{Z_r=4.0}^{5.0} \sum_{n=0}^{M-1} x(ADC_3)\exp[-j(\frac{2\pi f_0}{f_s} + \frac{2\pi f_L}{Mf_s}r)n] \\
R_2 = \sum_{Z_r=4.0}^{5.0} X(Z_r) = \sum_{Z_r=4.0}^{5.0} \sum_{n=0}^{M-1} x(ADC_4)\exp[-j(\frac{2\pi f_0}{f_s} + \frac{2\pi f_L}{Mf_s}r)n]
\end{cases}
\tag{4}
$$

where $ADC_1$, $ADC_2$ are the voltage obtained from the 2 concentration detection circuits, $ADC_3$, $ADC_4$ are the voltage obtained from the 2 reference channel circuits.

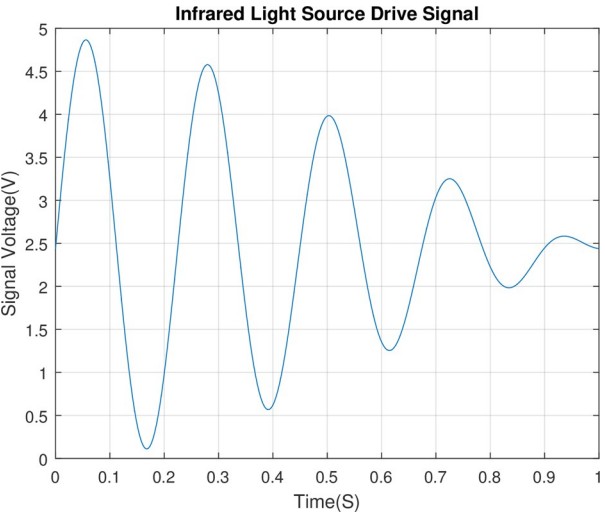

**Fig 6. Infrared light source drive voltage signal timing diagram.**

As such, 4 ratios of the modulus sums, Two infrared pyroelectric 3.4$\mu$m bands(methane detection bands) are compared to two 3.91$\mu$m bands (reference bands), that are calculated by the CZT algorithm in the characteristic frequency band can be represented as follow:

$$\begin{cases} Q_1 = \dfrac{M_1}{R_1} \\ Q_2 = \dfrac{M_1}{R_2} \\ Q_1 = \dfrac{M_2}{R_1} \\ Q_1 = \dfrac{M_2}{R_2} \end{cases} \tag{5}$$

where $Q_1$, $Q_2$, $Q_3$ and $Q_4$ are the ratio of modulus between 2 detection channels and 2 reference channels.

Then, combustible gas concentrations from 4 redundant combinations can be calculated as follows:

$$\begin{cases} C_1 = 1 - \dfrac{M_1}{R_1} \\ C_2 = 1 - \dfrac{M_1}{R_2} \\ C_1 = 1 - \dfrac{M_2}{R_1} \\ C_1 = 1 - \dfrac{M_2}{R_1} \end{cases} \tag{6}$$

where $C_1$, $C_2$, $C_3$ and $C_4$ are the combustible gas concentration from 4 redundant combinations.

Ultimately, the concentration of the combustible gas to be detected can be calculated as follow:

$$COL = \bar{Q} = \frac{C_1 + C_2 + C_3 + C_4}{4} \tag{7}$$

where $COL$ is the result of the combustible gas concentration to be detected, $\bar{Q}$ is the average of the concentrations of the four redundant combinations.

The trusted accuracy of the concentration can be evaluated by the concentration variance $S^2_{COL}$. Such as the trusted accuracy of the sensor is at the PPM level, when $S^2_{COL}$ is equal to 0.0000001.

$$S^2_{COL} = \frac{(\bar{Q} - C_1)^2 + (\bar{Q} - C_2)^2 + (\bar{Q} - C_3)^2 + (\bar{Q} - C_4)^2}{4} \tag{8}$$

## 4. Experiments and results

### 4.1 Combustible gas concentration calibration experiments and results

Methane gas from 0% to 90% LEL was produced by proportioning device of combustible gas concentration, as shown in Fig 7, and was used for combustible gas concentration calibration experiment to the sensor. Among them, the combustible gas with a concentration of 0% LEL

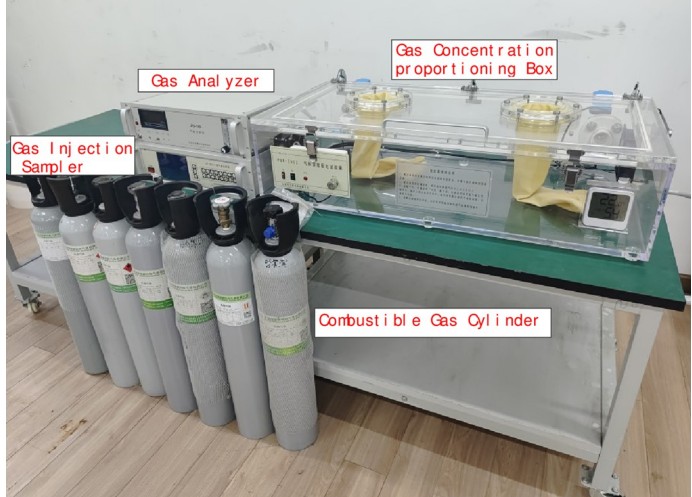

**Fig 7. Combustible gas concentration proportioning device diagram.**

was prepared by filling combustible gas concentration proportioning device with high purity air. The proportioning accuracy of this device is 0.0001PPM.

The results of the combustible gas concentration calibration experiment are analyzed as follows: First of all, from the 4 time domain signal figures (Figs 8–11), we can find that the signal strength of detection channels 1 and 2 decreased as the increase of the concentration of combustible gas (Figs 8 and 9), and that of the two reference channels remained the same all the time (Figs 10 and 11).

In the follow-up phase of the spectrum analysis to the data for 4 channels, it can be found that spectrum pecks were concentrated between 4-5Hz (Figs 12–15), Indicating that he characteristic frequency band was in this band. As well, the spectrum peaks of detection channels 1 and 2 decreased as the increase of concentration of combustible gas (Figs 14 and 15), and that of the two reference channels remained the same all the time (Figs 14 and 15). The same

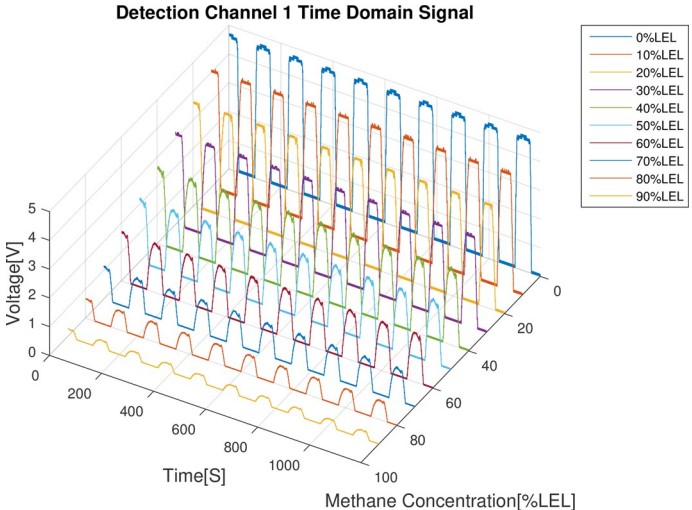

**Fig 8. Time domain results of detection channel 1.**

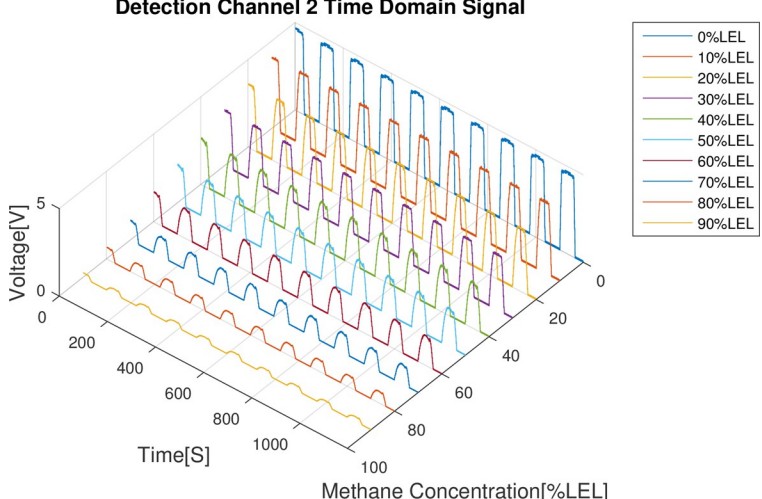

**Fig 9. Time domain results of detection channel 2.**

conclusion was obtained above when 4 channels were further analyzed in detail on the characteristic band (Figs 16–19).

In the final stage of the calculation of detection results for combustible gas from 0% to 100% LEL concentration, it can be seen that the results of each concentration of combustible gas were accurately calculated by the combustible gas concentration detection algorithm based on the CZT principle (Table 1). The maximum value of the concentration variance $S^2_{COL}$ of the detection result was 0.014PPM, and the minimum one was 0.0015PPM, indicating that the trusted accuracy of the concentration detection result can reach 0.014PPM level.

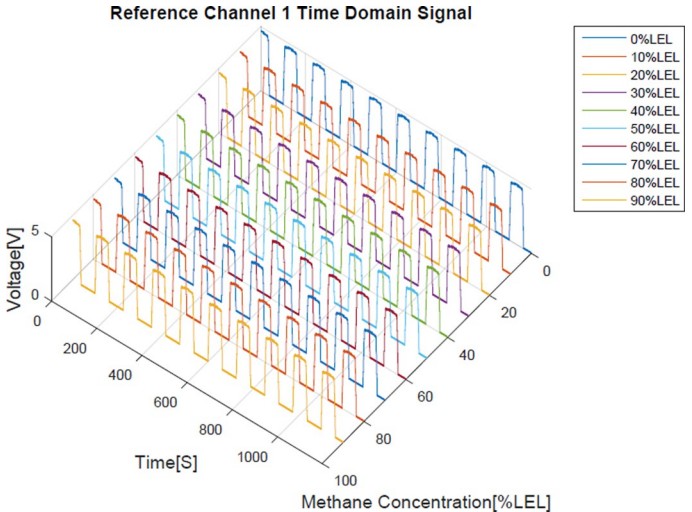

**Fig 10. Time domain results of reference channel 1.**

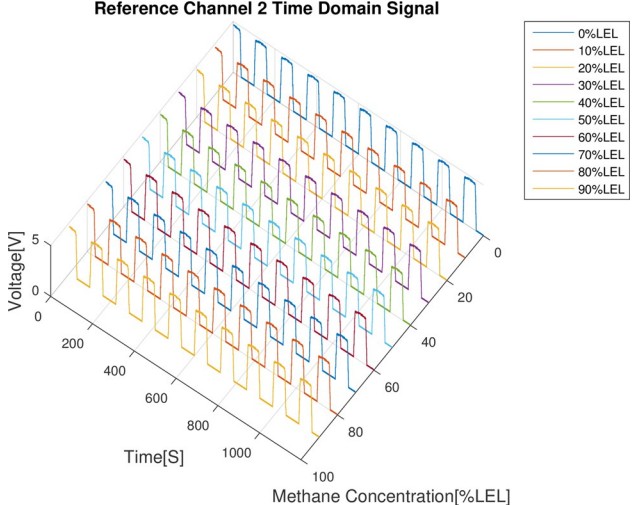

**Fig 11. Time domain results of reference channel 2.**

## 4.2 Combustible gas concentration limit detection experiments and results

Methane gas of the 0.5PPM concentration was produced by proportioning device of combustible gas concentration, and was used for the concentration limit detection experiment to the sensor. The time domain signal is as shown in Figs 20 and 21.

As shown in Table 2, the result of the concentration, which was calculated by the combustible gas concentration detection algorithm based on the CZT principle, was 0.5028PPM, and the concentration variance $S_{COL}^2$ was 0.015PPM. These data (in Table 2) are averages of three assay experiments with the same concentration. The concentration detection result of 0.5PPM can be obtained by rounding off the valid decimal places of the result 0.5028PPM.

It indicates that the trusted accuracy of the concentration detection result is 0.01PPM at least. Therefore, the concentration detection accuracy of the sensor can attain 0.01PPM.

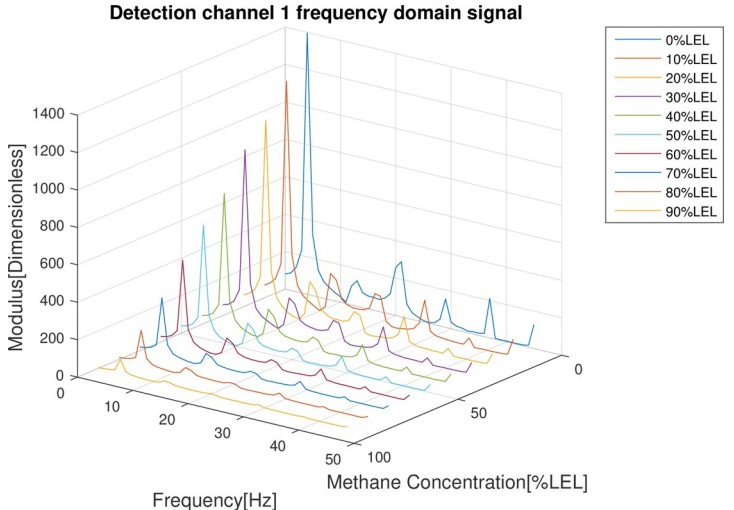

**Fig 12. Spectrum distribution of detection channel 1.**

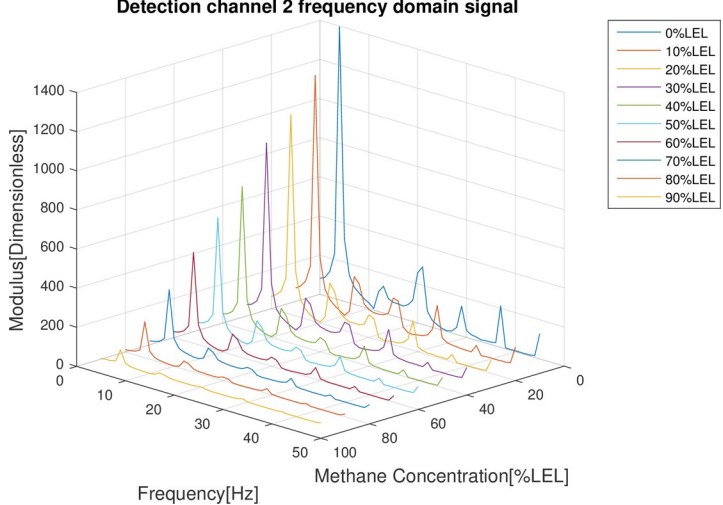

**Fig 13. Spectrum distribution of detection channel 2.**

To better test the sensor's effectiveness in detecting different concentrations of methane gas under different working conditions, methane gas with concentrations of 30%LEL and 70% LEL, 20%LEL and 60%LEL, and 40%LEL and 80%LEL were selected for the experiments in sections 4.3, 4.4 and 4.5, respectively. This also ensures that each set of experiments has a methane concentration below and above 50% LEL.

## 4.3 Anti-interference capability simulation experiments and results

The option incident window of the sensitive element was taped with the designed shading film (as shown in Fig 22) so that it was simulated that this window was attached by contaminants (such as condensation fog and dust particles). The material of the shading film is a designed black polycondensation resin polarizer film and is randomly opened with several small holes that are 2/3 of the total area on it. In this way, there will be 1/3 infrared light that is incident of

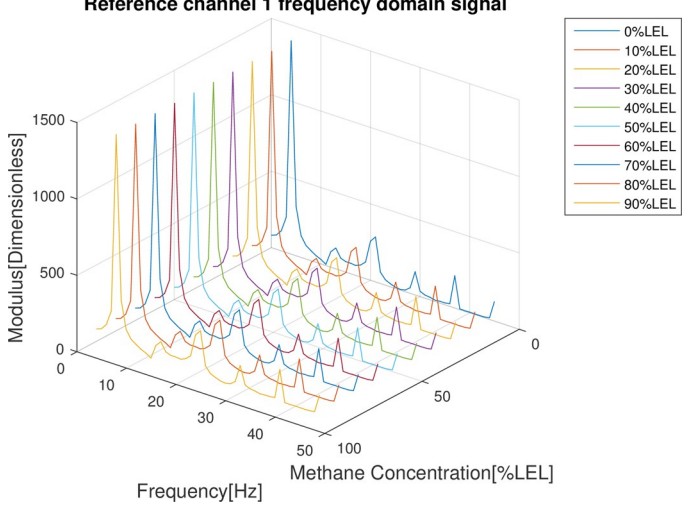

**Fig 14. Spectrum distribution of reference channel 1.**

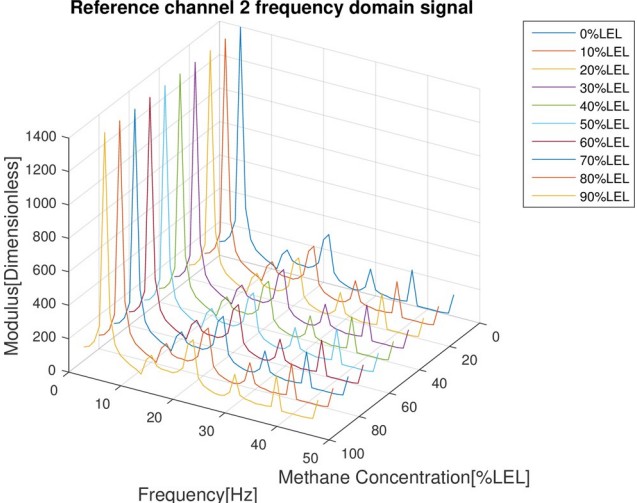

**Fig 15. Spectrum distribution of reference channel 2.**

the sensitive element, and another 2/3 of the infrared light will be blocked from the sensitive element by the black overshadow film, when the sensitive element is injected by the infrared light.

Methane gas of the 30%LEL and 70%LEL concentration was produced by proportioning device of combustible gas concentration, and was used for the simulation experiment of anti-interference ability to the sensor after being taped with the black shading film.

The data of the comparison of 4 groups (Table 3) demonstrated that the two detection results of two concentrations were identical before and after film application, and were 30% LEL and 40%LEL respectively. And there was a tiny difference in the trusted accuracy of two detection results. However, the trusted accuracy can always reach 0.01PPM. In the two concentration treatment groups, the modulus of 4 infrared signal detection channels were reduced by 2/3 after applying the black shading film.

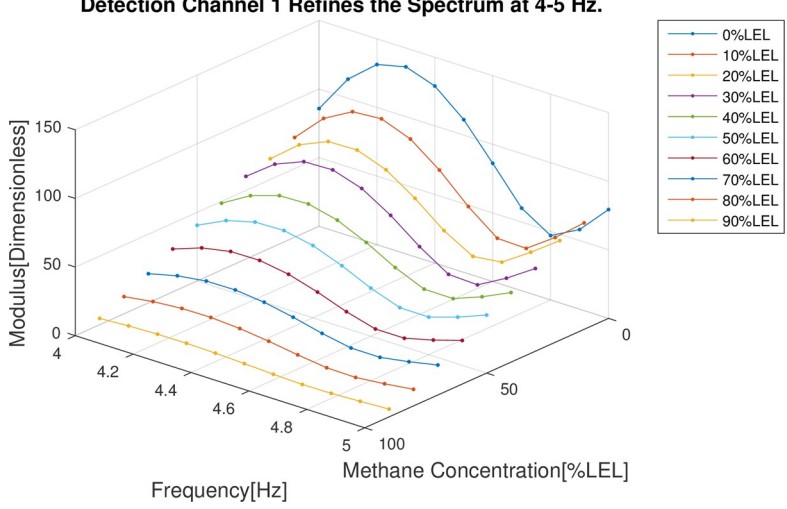

**Fig 16. 4–5 Hz frequency domain refinement results of detection channel 1.**

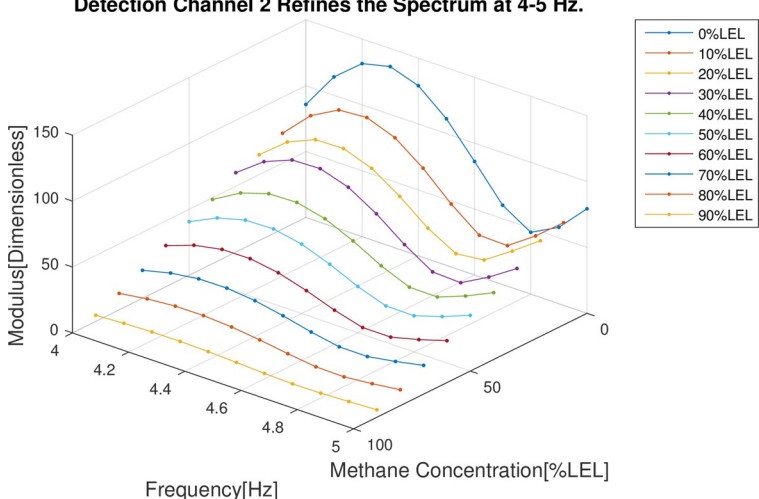

**Fig 17. 4–5 Hz frequency domain refinement results of detection channel 2.**

## 4.4 High humidity environment simulation experiments and results

Placed the sensor in the experiment box of the high humidity/concentration dust test device (Fig 23), and set the humidity of the device at 85% to perform the high humidity environment simulation experiment on the sensor. Opened the gas suction value to inject the combustible gas with the concentration of 20%LEL and 60%LEL into it respectively, when the humidity in the box had stabilized.

As shown in Table 4, the detection results of these two concentration were 20%LEL and 60%LEL, and the trusted accuracy can still reach 0.012PPM and 0.0061PPM in a high humidity environment of 85%. There was only a small reduction in the modulus of the 4 infrared signal detection channels.

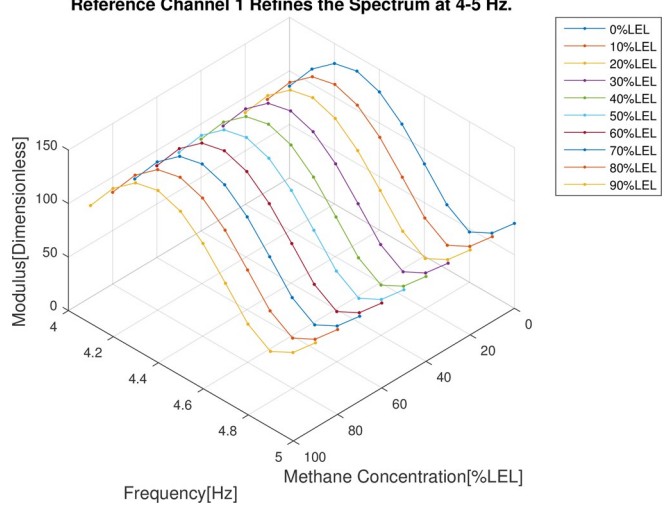

**Fig 18. 4–5 Hz frequency domain refinement results of reference channel 1.**

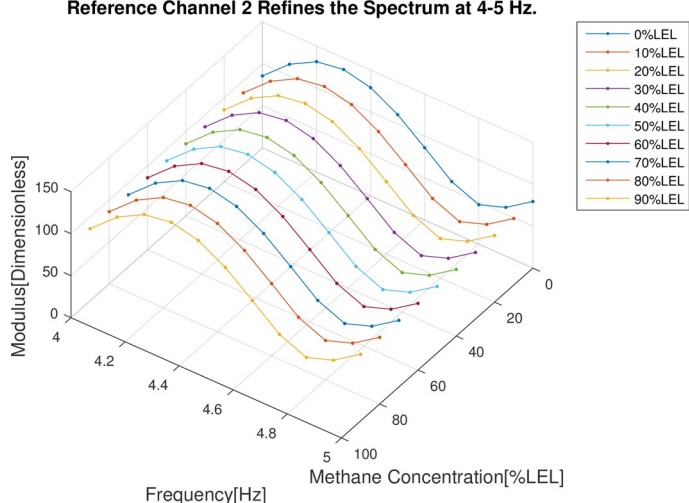

**Fig 19. 4–5 Hz frequency domain refinement results of reference channel 2.**

## 4.5 High concentration dust environment simulation experiments and results

Similarly, placed the sensor in the experiment box of the high humidity/concentration dust test device, and set the dust concentration to 100 mg/m$^3$(To test the detection capability of the sensor in the harshest industrial environments, the dust concentration was set to 100 mg/m3, because, the dust concentration will not exceed 100 mg/m3, even in harsh industrial environments such as mines.) in order that this dust concentration was used for the high concentration dust environment simulation experiment. Injected the combustible gas with the concentration of 40%LEL and 80%LEL into it, when the dust concentration in the box had settled down.

As shown in Table 5, the detection result of this two concentration were 40%LEL and 80% LEL, and the trusted accuracy can still reach 0.0063PPM and 0.0021PPM in a high concentration dust environment of 100 mg/m$^3$. There was only a small reduction in the modulus of the 4 infrared signal detection channels.

**Table 1. Calculated combustible gas concentration detection results for 0 ∼ 90% LEL methane concentration.**

| Concentration (%LEL) | $M_1$ | $M_2$ | $R_1$ | $R_2$ | $COL$ | $S^2_{COL}(PPM)$ |
|---|---|---|---|---|---|---|
| 0 | 1037.263935 | 1037.263932 | 1037.263937 | 1037.263936 | 0 | - |
| 10 | 933.537542 | 933.537539 | 1037.263937 | 1037.263936 | 0.1 | 0.0140 |
| 20 | 829.811148 | 829.811146 | 1037.263937 | 1037.263936 | 0.2 | 0.0120 |
| 30 | 726.084755 | 726.084752 | 1037.263937 | 1037.263936 | 0.3 | 0.0110 |
| 40 | 622.358361 | 622.358359 | 1037.263937 | 1037.263936 | 0.4 | 0.0090 |
| 50 | 518.631968 | 518.631966 | 1037.263937 | 1037.263936 | 0.5 | 0.0076 |
| 60 | 414.905574 | 414.905573 | 1037.263937 | 1037.263936 | 0.6 | 0.0061 |
| 70 | 311.179181 | 311.17918 | 1037.263937 | 1037.263936 | 0.7 | 0.0046 |
| 80 | 207.452787 | 207.452786 | 1037.263937 | 1037.263936 | 0.8 | 0.0030 |
| 90 | 103.726394 | 103.726393 | 1037.263937 | 1037.263936 | 0.9 | 0.0015 |

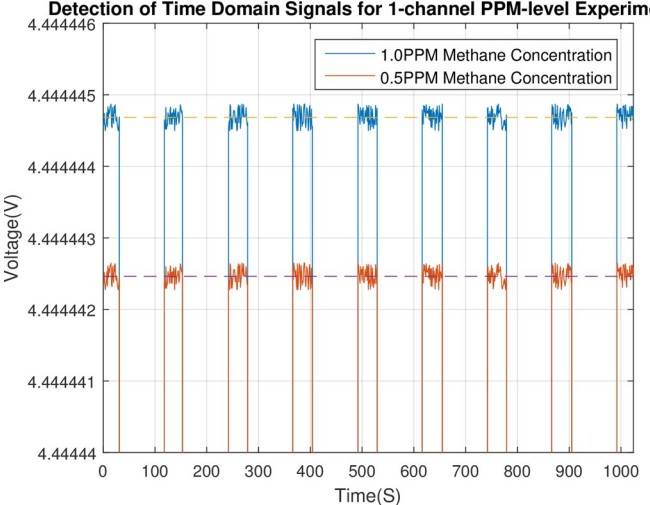

**Fig 20. Detection channel 1 time domain signal of the combustible gas detection channel 1 and 2 at methane concentration of 0.5PPM and 1PPM.**

## 5 Conclusions

1. The developed sensor can effectively detect the combustible gas from 0%LEL to 90%LEL, and the trusted accuracy of the detection result can reach 0.014PPM. This sensor can effectively detect the combustible gas at each concentration with high accuracy. Meanwhile, the method of multi-channel redundancy contributes to the improvement of the sensor reliability to a certain degree.

2. The limit of the measurement concentration of this detector can reach 0.5PPM, and the trusted accuracy is 0.01PPM. The design of the optical path structure of the sensitive element improves the sensor sensitivity so that it enables the effective detection of combustible gas with less than the PPM level.

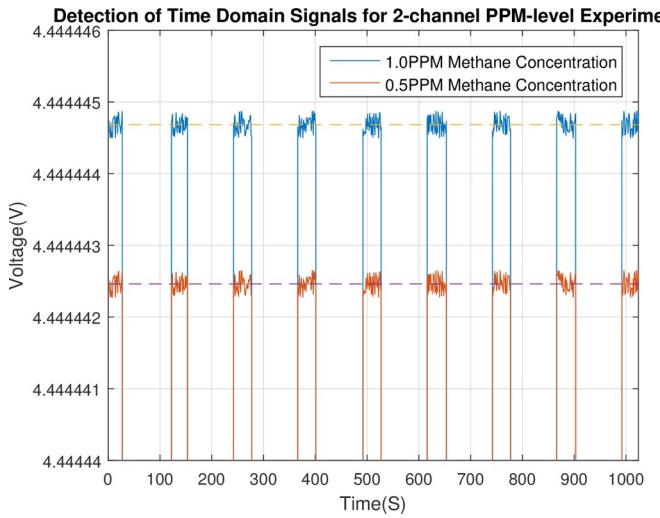

**Fig 21. Detection channel 2 time domain signal of the combustible gas detection channel 1 and 2 at methane concentration of 0.5PPM and 1PPM.**

**Table 2. Combustible gas concentration detection limit test results of 0.5PPM methane.**

| $M_1$ | $M_2$ | $R_1$ | $R_2$ | COL | $S^2_{col}/PPM$ |
|---|---|---|---|---|---|
| 1037.263401 | 1037.263451 | 1037.26395 | 1037.26394 | 0.5028PPM | 0.015 |

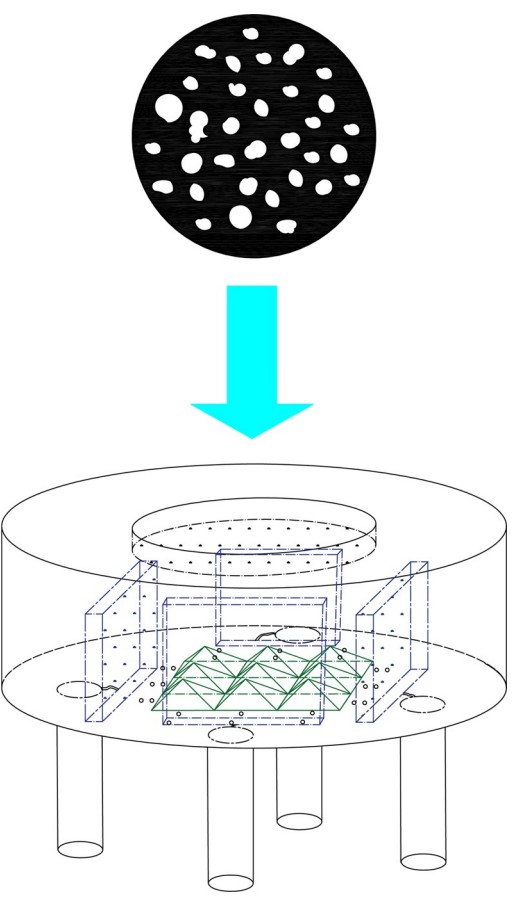

**Fig 22. Installation diagram of the shading film.**

**Table 3. Sensor anti-interference experimental calculation results.**

| Concentration (%LEL) | 30 Pre-film data | 30 Post-film data | 70 Pre-film data | 70 Post-film data |
|---|---|---|---|---|
| $M_1$ | 726.08 | 484.05 | 311.17 | 276.6 |
| $M_2$ | 726.08 | 484.05 | 311.17 | 276.6 |
| $R_1$ | 1037.2 | 691.5 | 1037.2 | 691.5 |
| $R_2$ | 1037.2 | 691.5 | 1037.2 | 691.5 |
| COL | 0.3 | 0.3 | 0.7 | 0.7 |
| $S^2_{COL}(PPM)$ | 0.0110 | 0.0186 | 0.0046 | 0.0071 |

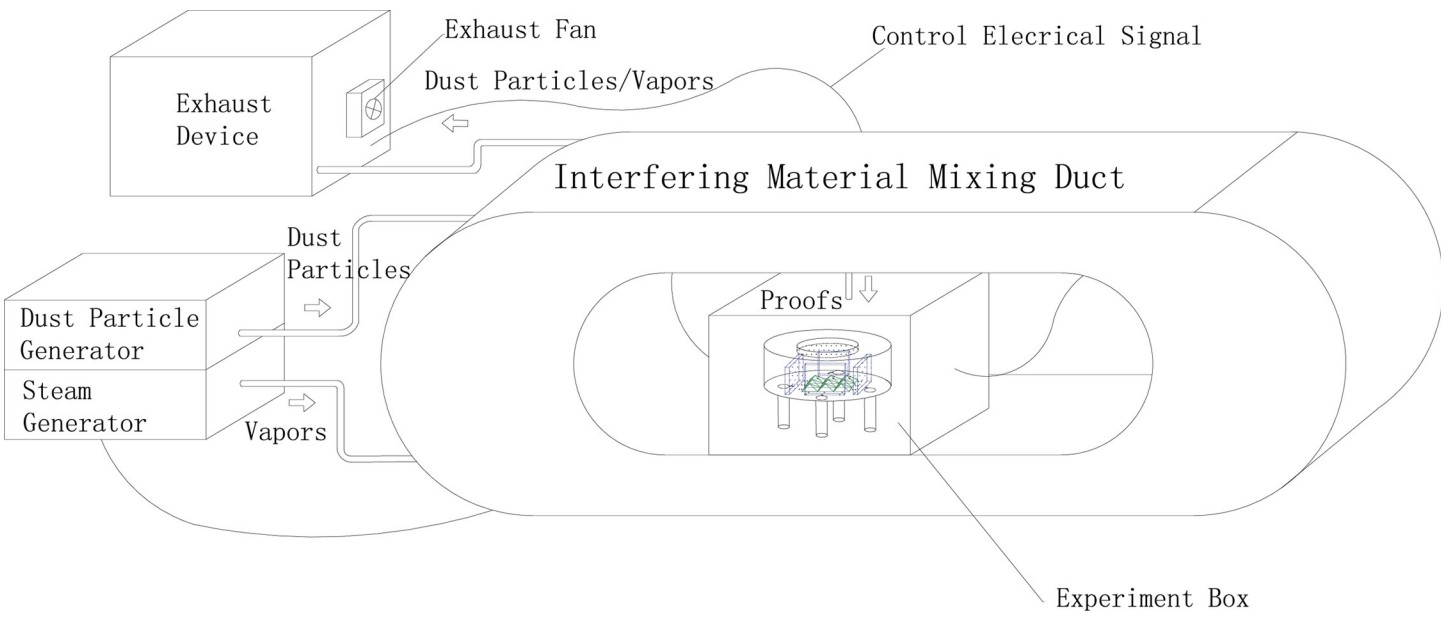

**Fig 23. High humidity / high concentration of dust test device schematic.**

**Table 4. Experimental results of the sensor working in high humidity environment.**

| Concentration (%LEL) | 20 | 60 |
|---|---|---|
| $M_1$ | 775.043612 | 387.52180 |
| $M_2$ | 775.043610 | 387.52180 |
| $R_1$ | 968.804517 | 968.80450 |
| $R_2$ | 968.804516 | 968.80450 |
| COL | 0.20000 | 0.60000 |
| $S^2_{COL}(PPM)$ | 0.012 | 0.0061 |

**Table 5. Experimental results of the sensor working in high concentration dust environment.**

| Concentration (%LEL) | 40 | 80 |
|---|---|---|
| $M_1$ | 458.0557537 | 150.4033 |
| $M_2$ | 458.0557524 | 150.4033 |
| $R_1$ | 763.4262576 | 752.0164 |
| $R_2$ | 763.4262569 | 752.0164 |
| COL | 0.400000001 | 0.8000 |
| $S^2_{COL}$ (PPM) | 0.0063 | 0.0021 |

3. The sensor can be still operational, and the trusted accuracy of detection results can still reach 0.01PPM under unfavorable conditions with 2/3 of the option incident window of the sensitive element blocked, humidity of 85%, and dust concentration of 100 mg/m3. The pyramidal beam splitter structure can improve sensor reliability, so that it can neutralize the effect of the optical window attached by contaminants.

4. Because of the 3.4um band selection for the detection channel, the sensor is only suitable for methane gas concentration detection.

## Supporting information

**S1 Data.**
(XLSX)

## Author Contributions

**Conceptualization:** Boqiang Wang.

**Software:** Boqiang Wang.

**Supervision:** Xuezeng Zhao, Yiyong Zhang.

**Writing – original draft:** Yiyong Zhang, Zhuogang Wang.

**Writing – review & editing:** Boqiang Wang, Xuezeng Zhao.

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
