## [Decision Letter · Decision Letter 0]

28 Nov 2023

PONE-D-23-32114Research on high performance combustible gas concentration sensor based on pyramid beam splitter matrixPLOS ONE

Dear Dr. wang,

Thank you for submitting your manuscript to PLOS ONE. After careful consideration, we feel that it has merit but does not fully meet PLOS ONE’s publication criteria as it currently stands. Therefore, we invite you to submit a revised version of the manuscript that addresses the points raised during the review process.

We look forward to receiving your revised manuscript.

Kind regards,

Yogendra Kumar Mishra, Ph. D.

Academic Editor

PLOS ONE

Journal Requirements:

4. Please ensure that you refer to Figure 11 and 12 in your text as, if accepted, production will need this reference to link the reader to the figure.

**Additional Editor Comments:**

Based on the referee comments, I suggest you to submit a carefully revised manuscript.

Reviewers' comments:

Reviewer's Responses to Questions

**Comments to the Author**

1. Is the manuscript technically sound, and do the data support the conclusions?

Reviewer #1: Yes

Reviewer #2: Yes

2. Has the statistical analysis been performed appropriately and rigorously? 

Reviewer #1: Yes

Reviewer #2: Yes

3. Have the authors made all data underlying the findings in their manuscript fully available?

Reviewer #1: Yes

Reviewer #2: Yes

4. Is the manuscript presented in an intelligible fashion and written in standard English?

Reviewer #1: Yes

Reviewer #2: Yes

5. Review Comments to the Author

Reviewer #1: Authors have designed a gas sensor with accuracy >0.01PPM without much change under 2/3rd window blocking, 85% humidity and dust levels of 100mg/m3. Overall, the paper is well organized and flows well. But some sections need to be revised to provide more insight and cut down on unnecessary details. Therefore, I would like to recommend the publication of this work in Materials Science and Engineering: B after some minor revisions. The details are listed as follows:

Comment 1: Introduction: Suggest adding more background on sensor construction and providing the reader a broader overview before jumping into the problem.

Comment 2: Introduction: Can the authors please add more details explaining the gap in the current research and describe the reasoning behind choosing pyramidal beam splitter matrix?

Comment 3: Section 2.1: It is unclear why dust or other contaminants do not affect the sensitivity of this sensor. Can the authors please add clarification explaining this point?

Comment 4: Suggest reducing the number of figures and only focusing on data critical to convey the key messages. For example: Consider keeping results either from reference channel 1 or from reference channel 2, instead of both of the reference channels.

Comment 5: Conclusions: Please be more specific and add quantification for the improvement of accuracy and reliability.

Reviewer #2: Overall Comments:

•The paper does a good work in describing the setup, experiments and results.

• Formatting and font is not consistent and should be made consistent.

1 Introduction

The introduction mentions the problem statement targeted and motivation to pursue a system with better accuracy and sensitivity. The following addition will make the introduction more robust:

• Authors should add typical values of sensor accuracy and sensitivity observed to provide readers a reference to compare the current work’s results. Appropriate citations should be added for the claimed values.

2 Design of sensitive element and measurement circuit

This section explains the fundamentals of the sensor principle, hardware and circuit well. The following addition will help further improve the quality of the work:

• The authors should mention that the calculations are discussed in the subsequent section to provide context to the readers for the line 114-116.

3 Combustible gas concentration detection algorithm based on CZT principle

This section does a good work in introducing the calculations used to determine the sensi- tivity and accuracy of the device. The following addition should be made:

• Add citation for line 145-146: ”CZT algorithm is often used for spectrum refinement in the characteristic bands of the signal, 146 and it has the advantages of flexible refinement scale and high accuracy.”

• The authors mention in line 154-155 ”We can derive equation (2) from equation (3).” It seems it should be the other way. Please clarify or modify accordingly.

4 Experiments and results

• Authors should clarify on line 262 ”The maximum value of the concentration variance of S2 the detection result was 0.014PPM, and the minimum one was 0.0015PPM, indicating that the trusted accuracy of the concentration detection result can reach 0.014PPM level in the most adverse condition.” what is meant by most adverse condition here and how has this experiment demonstrated this.

• For figure 11, authors should mention in the caption that the figure also includes data for 1 ppm methane.

• For the table 2 the authors should clarify in the caption that the data is only for 0.5 ppm methane.

• In section 4.2 The authors should comment on the delta between the expected concentration of the methane 5.0 ppm and the obtained values of 5.0380 ppm. The authors should additionally clarify how many times was this data collection performed. (Optional: Adding repeatability and reproducibility experiments would add robustness to the work).

• The authors should clarify why the tested concentrations in section 4.3, 4.4 and 4.5 were specifically 30/70, 20/60 and 40/80 respectively.

• Table 4 column mentions concentration of 20 twice. Correct one of them to 60 appropriately.

• The authors should Clarify why the dust concentration was chosen as 100 mg/m3. Mention how this emulates a real world situation.

Conclusion

The conclusion provides a high level summary of the key insights from the work. However the conclusion should be more quantitative. The authors should also clarify if this conclusion can be translated for other types of gases as well since the current work focuses on methane. If so the authors should clarify why.

6. PLOS authors have the option to publish the peer review history of their article (what does this mean?). If published, this will include your full peer review and any attached files.

Reviewer #1: No

Reviewer #2: No

---

## [Author Response · Author response to Decision Letter 0]

20 Dec 2023

Reviewer #1: Authors have designed a gas sensor with accuracy >0.01PPM without much change under 2/3rd window blocking, 85% humidity and dust levels of 100mg/m3. Overall, the paper is well organized and flows well. But some sections need to be revised to provide more insight and cut down on unnecessary details. Therefore, I would like to recommend the publication of this work in Materials Science and Engineering: B after some minor revisions. The details are listed as follows:

Comment 1: Introduction: Suggest adding more background on sensor construction and providing the reader a broader overview before jumping into the problem.

Response: Thank you for pointing this out. We agree with this comment. Therefore, we have added the following in section 1.

“Non-dispersive infrared (NDIR) is the most widely used sensor for combustible gas concentration. Figure 1 shows a schematic diagram of a simple NDIR gas sensor. Typically, emission from a broadband source is passed through two filters, one covering the whole absorption band of the target gas (in the active channel), and the other covering a neighboring non-absorbed region (the reference channel). Provided that the chosen active and reference channel filters do not overlap significantly with the absorption bands of other gas species present in the application, cross-sensitivity to other gases lies below the limit of detection. Detects the concentration of combustible gas by the degree of absorption of a light source.

Fig1. Schematic structure of NDIR combustible gas concentration sensor”

Comment 2: Introduction: Can the authors please add more details explaining the gap in the current research and describe the reasoning behind choosing pyramidal beam splitter matrix?

Response: Thank you for pointing this out. We agree with this comment. Therefore, we have added the following in section 1.

“The previous studies did not have a precise design of the optical path structure, which resulted in the infrared light not being sufficiently absorbed by the combustible gas in the tiny sensor optical path structure. Thus, the sensitivity and accuracy of the sensor could not meet the requirements. The pyramid beam splitter creates multiple reflections in the tiny sensor optical path structure, greatly increasing the effective optical range and allowing infrared light to be fully absorbed by combustible gases.”

Comment 3: Section 2.1: It is unclear why dust or other contaminants do not affect the sensitivity of this sensor. Can the authors please add clarification explaining this point?

Response: Thank you for pointing this out. We agree with this comment. Therefore, we have added the following in section 2.1.

 “Through the reflection of the pyramid reflector matrix in the internal optical structure of the sensor, the infrared light can be optically mixed, so that each channel receives the same infrared light composition. This process also increases the effective optical range, so that the sensor's sensitivity and detection accuracy are not affected by the adherence of contaminants.”

Comment 4: Suggest reducing the number of figures and only focusing on data critical to convey the key messages. For example: Consider keeping results either from reference channel 1 or from reference channel 2, instead of both of the reference channels.

Response: Thank you for pointing this out. We agree with this comment. Therefore, We deleted the C1-C4 concentration data in Table 1, and the data for Q1-Q2 and C1-C4 in Table 2, and the data for C1-C4 in Table 3, and the data for Q1-Q2 in Table 4, and the data for Q1-Q2 in Table 5.

The data of detection channels 1,2 and reference channels 1, 2 retained would be more intuitive and therefore retained.

Comment 5: Conclusions: Please be more specific and add quantification for the improvement of accuracy and reliability.

Response: Thank you for pointing this out. We agree with this comment. Therefore, we have added followings in section 5.

“(1) The developed sensor can effectively detect the combustible gas from 0%LEL to 90%LEL, and the trusted accuracy of the detection result can reach 0.014PPM.…”

“(2) The limit of the measurement concentration of this detector can reach 0.5PPM, and the trusted accuracy is 0.01PPM.…”

“(3) The sensor can be still operational, and the trusted accuracy of detection results can still reach 0.01PPM under unfavorable conditions with 2/3 of the option incident window of the sensitive element blocked, humidity of 85%, and dust concentration of 100 mg/m3.…”

Reviewer #2: Overall Comments:

•The paper does a good work in describing the setup, experiments and results.

• Formatting and font is not consistent and should be made consistent.

Response: Formatting and fonts have been adjusted.

1 Introduction

The introduction mentions the problem statement targeted and motivation to pursue a system with better accuracy and sensitivity. The following addition will make the introduction more robust: • Authors should add typical values of sensor accuracy and sensitivity observed to provide readers a reference to compare the current work’s results. Appropriate citations should be added for the claimed values.

Response: Thank you for pointing this out. We agree with this comment. Therefore, we have added the following in section 1.

 “In the very early stages of a fire, a sensor with at least 3 PPM sensitivity and 1 PPM accuracy is required [8]” 

And we have added the reference [8]” Park G, Lyu G, Jo Y D, et al. A Study on the Development and Accuracy Improvement of an IR Combustible Gas Leak Detector with Explosion Proof[J]. Journal of The Korean Institute of Gas, 2014, 18(3): 1-12.”

2 Design of sensitive element and measurement circuit

This section explains the fundamentals of the sensor principle, hardware and circuit well. The following addition will help further improve the quality of the work:

• The authors should mention that the calculations are discussed in the subsequent section to provide context to the readers for the line 114-116.

Response: Thank you for pointing this out. We agree with this comment. Therefore, we have added the following in section 2.2.

” The calculation of combustible gas concentration is discussed in section 3.”

And “The electrical signal difference can be adjusted by adjusting the position of the digital potentiometer Wiper U25 in Figure 4.”

 3 Combustible gas concentration detection algorithm based on CZT principle

This section does a good work in introducing the calculations used to determine the sensitivity and accuracy of the device. The following addition should be made:

• Add citation for line 145-146: ”CZT algorithm is often used for spectrum refinement in the characteristic bands of the signal, 146 and it has the advantages of flexible refinement scale and high accuracy.”

Response: Thank you for pointing this out. We agree with this comment. Reference [16] has been added: “Nudi F, Iskandrian A E, Schillaci O, et al. Diagnostic accuracy of myocardial perfusion imaging with CZT technology: systemic review and meta-analysis of comparison with invasive coronary angiography[J]. JACC: Cardiovascular Imaging, 2017, 10(7): 787-794.”

• The authors mention in line 154-155 ”We can derive equation (2) from equation (3).” It seems it should be the other way. Please clarify or modify accordingly.

Response: Thank you for pointing this out. We agree with this comment. Therefore, we have changed the text from” We can derive equation (2) from equation (3)” to” We can derive equation (3) from equation (2)”.

4 Experiments and results

• Authors should clarify on line 262 ”The maximum value of the concentration variance of S2 the detection result was 0.014PPM, and the minimum one was 0.0015PPM, indicating that the trusted accuracy of the concentration detection result can reach 0.014PPM level in the most adverse condition.” what is meant by most adverse condition here and how has this experiment demonstrated this.

Response: Thank you for pointing this out. The words "in the most adverse condition." have been deleted, due to a clerical error.

• For figure 11, authors should mention in the caption that the figure also includes data for 1 ppm methane.

Response: Thank you for pointing this out. We agree with this comment. Therefore, we have changed the caption of Figure 11 from” The time domain signal of the combustible gas detection channel 1 and 2 at 0.5PPM methane concentration” to” The time domain signal of the combustible gas detection channel 1 and 2 at methane concentration of 0.5PPM and 1PPM”

• For the table 2 the authors should clarify in the caption that the data is only for 0.5 ppm methane.

Response: Thank you for pointing this out. We agree with this comment. Therefore, we have changed the text from” Table 2. Combustible gas concentration detection limit test results” to” Table 2. Combustible gas concentration detection limit test results of 0.5PPM methane”.

• In section 4.2 The authors should comment on the delta between the expected concentration of the methane 5.0 ppm and the obtained values of 5.0380 ppm. The authors should additionally clarify how many times was this data collection performed. (Optional: Adding repeatability and reproducibility experiments would add robustness to the work).

Response: There was a clerical error in the data in table 2, which has been corrected.

Thank you for pointing this out. We agree with this comment. Therefore, we have added the following in section 4.2.

“ These data (in Table 2) are averages of three assay experiments with the same concentration. The concentration detection result of 0.5PPM can be obtained by rounding off the valid decimal places of the result 0.5028PPM.”

• The authors should clarify why the tested concentrations in section 4.3, 4.4 and 4.5 were specifically 30/70, 20/60 and 40/80 respectively.

Response: Thank you for pointing this out. We agree with this comment. Therefore, Therefore, we have added the following in section 4.2.

“To better test the sensor's effectiveness in detecting different concentrations of methane gas under different working conditions, methane gas with concentrations of 30%LEL and 70%LEL, 20%LEL and 60%LEL, and 40%LEL and 80%LEL were selected for the experiments in sections 4.3, 4.4 and 4.5, respectively. This also ensures that each set of experiments has a methane concentration below and above 50% LEL.”

• Table 4 column mentions concentration of 20 twice. Correct one of them to 60 appropriately.

Response: Thank you for pointing this out. We agree with this comment. Therefore, we have changed the text from” 20” to” 60”.

• The authors should Clarify why the dust concentration was chosen as 100 mg/m3. Mention how this emulates a real world situation.

Response: Thank you for pointing this out. We agree with this comment. Therefore, we have added the following in section 4.5.

“To test the detection capability of the sensor in the harshest industrial environments, the dust concentration was set to 100 mg/m3, because, the dust concentration will not exceed 100 mg/m3, even in harsh industrial environments such as mines..”

Conclusion

The conclusion provides a high level summary of the key insights from the work. However the conclusion should be more quantitative. The authors should also clarify if this conclusion can be translated for other types of gases as well since the current work focuses on methane. If so the authors should clarify why.

Response: Thank you for pointing this out. We agree with this comment. Therefore, we have added followings in section 5.

“(1) The developed sensor can effectively detect the combustible gas from 0%LEL to 90%LEL, and the trusted accuracy of the detection result can reach 0.014PPM.…”

“(2) T The limit of the measurement concentration of this detector can reach 0.5PPM, and the trusted accuracy is 0.01PPM.…”

“(3) The sensor can be still operational, and the trusted accuracy of detection results can still reach 0.01PPM under unfavorable conditions with 2/3 of the option incident window of the sensitive element blocked, humidity of 85%, and dust concentration of 100 mg/m3.…”

“(4)Because of the 3.4um band selection for the detection channel, the sensor is only suitable for methane gas concentration detection.”

---

## [Decision Letter · Decision Letter 1]

27 Feb 2024

Research on high performance combustible gas concentration sensor based on pyramid beam splitter matrix

PONE-D-23-32114R1

Dear Dr. wang,

We’re pleased to inform you that your manuscript has been judged scientifically suitable for publication and will be formally accepted for publication once it meets all outstanding technical requirements.

Kind regards,

Yogendra Kumar Mishra, Ph. D.

Academic Editor

PLOS ONE

Additional Editor Comments (optional):

Reviewers' comments:

Reviewer's Responses to Questions

**Comments to the Author**

1. If the authors have adequately addressed your comments raised in a previous round of review and you feel that this manuscript is now acceptable for publication, you may indicate that here to bypass the “Comments to the Author” section, enter your conflict of interest statement in the “Confidential to Editor” section, and submit your "Accept" recommendation.

Reviewer #1: All comments have been addressed

Reviewer #2: All comments have been addressed

2. Is the manuscript technically sound, and do the data support the conclusions?

Reviewer #1: Yes

Reviewer #2: Yes

3. Has the statistical analysis been performed appropriately and rigorously? 

Reviewer #1: Yes

Reviewer #2: Yes

4. Have the authors made all data underlying the findings in their manuscript fully available?

Reviewer #1: Yes

Reviewer #2: Yes

5. Is the manuscript presented in an intelligible fashion and written in standard English?

Reviewer #1: Yes

Reviewer #2: Yes

6. Review Comments to the Author

Reviewer #1: The authors have answered all the questions raised and included appropriate revisions in the manuscript. I recommend the publication of this paper in PLOS ONE.

Reviewer #2: (No Response)

7. PLOS authors have the option to publish the peer review history of their article (what does this mean?). If published, this will include your full peer review and any attached files.

Reviewer #1: No

Reviewer #2: No
